# Acid Mesoporous Carbon Monoliths from Lignocellulosic Biomass Waste for Methanol Dehydration

**DOI:** 10.3390/ma12152394

**Published:** 2019-07-26

**Authors:** Paul O. Ibeh, Francisco J. García-Mateos, Ramiro Ruiz-Rosas, Juana María Rosas, José Rodríguez-Mirasol, Tomás Cordero

**Affiliations:** Departamento de Ingeniería Química, Campus de Teatinos s/n, Universidad de Málaga, 29010 Málaga, Spain

**Keywords:** activated carbon monolith, lignin, biomass waste, acid catalyst, methanol dehydration

## Abstract

Activated carbon monoliths (ACMs), with 25 cells/cm^2^, were prepared from the direct extrusion of Alcell, Kraft lignin and olives stones particles that were impregnated with phosphoric acid, followed by activation at 700 °C. These ACMs were used as catalysts for methanol dehydration reaction under air atmosphere. ACM that was prepared from olive stone and at impregnation ratio of 2, OS2, showed the highest catalytic activity, with a methanol conversion of 75%, a selectivity to dimethyl ether (DME) higher than 90%, and a great stability under the operating conditions studied. The results suggest that the monolithic conformation, with a density channel of 25 cells/cm^2^ avoid the blockage of active sites by coke deposition to a large extent. Methanol conversion for OS2 was reduced to 29% in the presence of 8%v water, at 350 °C, although the selectivity to DME remained higher than 86%. A kinetic model of methanol dehydration in the presence of air was developed, while taking into account the competitive adsorption of water. A Langmuir-Hinshelwood mechanism, whose rate-limiting step was the surface reaction between two adsorbed methanol molecules, represented the experimental data under the conditions studied very well. An activation energy value of 92 kJ/mol for methanol dehydration reaction and adsorption enthalpies for methanol and water of −12 and −35 kJ/mol, respectively, were obtained.

## 1. Introduction

Dimethyl ether (DME) is recognized as a clean diesel substitute, being environmental friendly with zero ozone depletion potential and a key intermediate for other important chemicals, such as olefins. DME can be produced by methanol dehydration reactions while using acid catalysts, such as zeolite materials, γ-Al_2_O_3_, and acid modified γ-Al_2_O_3_. However, most of these solid acid catalysts suffer from fast deactivation due to coke deposition over strong acid sites, and, in addition, their catalytic activities are usually negatively influenced by the presence of water [1,2]. Therefore, the look for alternative catalysts with improved levels of performance is considered to be an important challenge.

In this sense, the use of carbon materials as catalysts and/or catalyst supports is receiving great attention [3]. However, only few works analyzed the use of activated carbons for methanol dehydration due to their lowest surface acidity. This acidity can be eventually increased by different methods [4,5,6], but no outstanding results were obtained due to the low stability of the acidic surface sites produced. In this regard, our research group has reported the preparation and characterization of activated carbons from different lignocellulosic biomass waste by chemical activation with phosphoric acid at specific preparation conditions [7,8]. This preparation method provides activated carbons with a high development of the porosity, relatively high oxidation resistance [9,10], and a relatively large presence of low to moderate acid strength sites [11,12]. These activated carbons were used as catalyst supports [8,13,14], and even directly as acid catalysts for alcohols dehydration reaction [11,15,16,17], including methanol dehydration. In the absence of oxygen, this type of catalyst showed a gradual deactivation due to coke deposition. However, high stability and selectivity to DME was reached under the air atmosphere, due to the fact that oxygen prevented the acid carbon catalyst from coke deposition, without gasification of the carbon catalyst was noticeable [12].

Activated carbons are usually produced in a powder shape. However, a monolith conformation presents diverse advantages in catalytic applications, due to the fact that they offer better mass transfer, low pressure drop, thermal stability, and good mechanical strength [18]. One option to prepare activated carbon monoliths (ACMs) with structured channels is by zeolite templated carbons, followed by the removal of the zeolite with acid [19], although this method can produce templated structures that crack during preparation. Other option is to directly obtain ACM by the extrusion of a carbonaceous paste. In this sense, only a few works reported the preparation of ACMs with a honeycomb structure by this method [20,21]. To our best knowledge, the preparation of ACM with well-defined axial channels by extrusion of lignin was firstly reported by our research group [22].

In this work, ACMs with axial channels were prepared from the direct extrusion of different lignocellulosic materials, Alcell, Kraft lignin and olives stones, impregnated with phosphoric acid, followed by activation at 700 °C. The activity and stability of these ACMs were analyzed for methanol dehydration reaction in air atmosphere. The influence of the presence of water on the dehydration of methanol were also studied. A kinetic model was also proposed to reproduce the experimental results.

## 2. Materials and Method

### 2.1. ACMs Preparation

The ACMs were prepared from different biomass raw materials: Alcell^®^ lignin (AL) (Repap Technologies Inc., Vancouver, Canada), eucalyptus Kraft lignin (KL) supplied by Empresa Nacional de Celulosas (ENCE, Pontevedra, Spain), and olive stone (OS), which were supplied by S.C.A. Olivarera y Frutera San Isidro, Periana (Málaga, Spain).

The precursors were impregnated with phosphoric acid at impregnation ratios (mass H_3_PO_4_/mass raw material) of 1 and 2, this last only for olive stone. The impregnated samples, dried for 24 h at 60 °C, were extruded at room temperature and 0.8 MPa. These monoliths were directly activated at 700 °C, under nitrogen flow, except in the case of lignin, for which a previous stabilization step was also required. Subsequently, the activated monoliths were washed with distilled water until constant pH in the eluate. A more detailed explanation about the preparation method of the ACMs can be found elsewhere [22].

The extruder, which was designed by our research group, mainly consisted of a cylindrical mould of stainless steel with an internal diameter of 2 cm. A die element with 19 pins was used for the preparation of these channeled ACMs, with the objective of obtaining monoliths with a cell density of 25 channels/cm^2^.

### 2.2. ACM Characterization

The morphology of the ACMs was analyzed by scanning electron microscopy (SEM), with a JEOL JSM-6490LV instrument (Tokyo, Japan), working at a high voltage of 20–25 kV.

The porosity of the ACMs was studied by N_2_ adsorption–desorption at −196 °C, while using a micrometrics instrument (ASAP 2020 model, Micromeritics, Norcross, GA, USA) and by Hg porosimetry (Autopore IV model, Micromeritics, Norcross, GA, USA). The ACMs were previously outgassed for 8 h at 150 °C. BET equation and the t-method were used for calculating the corresponding textural parameters from the N_2_ adsorption isotherm data.

The surface chemistry of the ACMs was studied by X-ray photoelectron spectroscopy (XPS) and temperature-programmed desorption (TPD). XPS analyses of the samples were obtained in a 5700C model Physical Electronics apparatus, with Mg Kα radiation (1253.6 eV, Physical Electronics, Feldkirchen, Germany). Temperature-programmed desorption (TPD) was used for evaluating the carbon-oxygen surface groups. The surface groups of acidic character (carboxylic, lactonic) evolve as CO_2_, whereas those of non-acid character (carbonyl, ether, quinone) and phenol groups give rise to CO. Anhydride surface groups evolve as both CO and CO_2_. The evolution of CO and CO_2_ as function of the temperature was obtained in a custom quartz fixed bed reactor placed inside an electrical furnace, by using non-dispersive infrared (NDIR) gas analyzers (Siemens ULTRAMAT 22, München, Germany). Milled and sieved ACMs were heated from room temperature to 930 °C, in N_2_ flow (heating rate of 10 °C/min.).

Acid-base characterization was carried out by analyzing dehydration-dehydrogenation of 2-propanol (IPA) as a model reaction test. The decomposition of IPA was performed under inert atmosphere and at atmospheric pressure in a quartz fixed bed microreactor (6 mm i.d.) that was placed inside a vertical furnace with temperature control. IPA was fed to the reactor by a syringe pump (Cole-Parmer^®^ 74900-00-05 model, Vernon Hills, IL, USA), ensuring a constant controlled IPA flow, using a partial pressure of IPA of 0.03 atm and a space time of W/F_0IPA_ = 0.1 g·s/µmol. The concentrations of reactant and products were analyzed by gas chromatography (490 micro-GC equipped with PPQ, 5A molsieve, and Wax columns, Agilent Technologies, Santa Clara, CA, USA).

### 2.3. Methanol Dehydration

The decomposition of methanol was performed in the same microreactor used for IPA decomposition test. The reaction was carried out with 200 mg of ACM, under air atmosphere, at temperatures between 150 and 375 °C. All lines were heated up to 120 °C in order to avoid any condensation. The inlet methanol partial pressure was 0.03 atm and the water partial pressures were varied from 0 to 0.08 atm, at a space time of 0.1 g·s/µmol.

The outlet gas concentrations were quantified by gas chromatography (490 micro-GC equipped with PPQ, 5A molsieve, and Wax columns, Agilent Technologies, Santa Clara, CA, USA).

The conversion was defined as the ratio of the amount of methanol that was converted to the amount of methanol added to the reactor. The selectivity (in mol%) was defined as the molar ratio of a specific product to the amount of methanol converted. The carbon balance was reached with an error lower than 3% in all cases.

## 3. Results and Discussion

### 3.1. ACMs Characterization

Cylindrical ACMs were obtained with preparations yields that ranged from 30 to 40% (ACMs mass/raw material mass). Figure 1 shows SEM micrographs of the ACMs obtained from olive stone, Alcell lignin and Kraft lignin, at an impregnation ratio of 1. The ACMs from olive stone and Alcell lignin presented a cross section diameter of approximately 0.7 and 0.8 cm, with cell densities of 54 and 38 channels/cm^2^, respectively (see Figure 1a,d). Details regarding the channels of both ACMs showed quite regular circular hollows along all the length of the monolith, with channel sizes ranging from 700 to 800 µm. In case of Kraft lignin derived activated carbon monolith, a more similar xerogel-like morphology was obtained, making the channels practically indiscernible, due to their collapse during the heat treatment at 700 °C. Accordingly, only discs were depicted in this figure.

These ACMs also present wall thickness from 250 to 800 μm, hydraulic diameter from 0.7 to 4 cm, open frontal area between 5% and 25%, and geometric surface area from 0.7 to 1 cm^2^/cm^3^. Furthermore, the compression strength values of these ACMs ranged from 4.12 to 7.56 MPa, with the highest value being obtained for the Alcell lignin one [22].

As can be seen, the preparation method here exposed allowed for obtaining circular shaped ACMs with axial channels, at very low pressure (0.8 MPa), in contrast to monolithic discs that were reported in the literature, which were obtained at 130 MPa [23,24]. Furthermore, most of these ACMs reported in the literature were usually made by compressing the already reported activated carbon in the presence of a binder, obtaining the corresponding ACM in the form of solid discs [25,26,27,28,29]. However, these binding components can produce a significant reduction in the porosity of the resulting monolith.

Other authors proposed the preparation of ACMs by the extrusion of the impregnated material with different activating agents, in a similar way to the one proposed in this work, but also with the goal of obtaining ACM discs [30,31,32,33,34,35]. In this sense, the preparation of ACMs with well-defined channels, like those here presented, was less reported and could be considered to be very useful for different applications.

Figure 2 collects a brief summary of some textural parameters for different ACMs that were derived from the N_2_ adsorption-desorption isotherms at −196 °C. The values of external surface area (A_t_) were considerably lower than those for the apparent surface area (A_BET_), which suggested that the main surface area is contained in micropores. The high values of A_BET_ that were found for the different ACMs, which ranged from 700 to 1500 m^2^/g, also indicate the high development of the porosity that was obtained in these monoliths. Specifically, the highest values for these textural parameters were obtained for the ACM derived from olive stone and prepared at the impregnation ratio of 2. These values are in the same range than those that were reported for a powder activated carbon from olive stone prepared by chemical activation with phosphoric acid at an impregnation ratio of 1, and a temperature of 500 °C (A_BET_ ~900 m^2^/g) [11], which suggests that the extrusion of the impregnated samples, followed by activation, does not limit the development of porosity, when compared to the traditional method that was carried out by the extrusion of a mixture of activated carbon with a binder. In any case, ACMs from both Alcell lignin and olive stone presented A_BET_ values that were higher than 1000 m^2^/g; values that were quite remarkable for catalytic applications.

Figure 3 shows the cumulative pore volume of the ACMs derived from both the N_2_ adsorption at −196 °C by using two-dimensional-Non Localized Density Functional Theory (2D-NLDFT) heterogeneous adsorption models for carbon slit-shaped pores and Hg porosimetry. Specifically, a cumulative pore volume at different pore size intervals is represented for the different catalysts. As can be seen, OS2 and AL1 presented a very similar volume of micropores (pore sizes lower than 2 nm), but OS2 was the ACM with the largest contribution of pore size, between 0 and 10 nm (in the range of micropores and narrow mesopores). In the interval of pore size between 1 × 10^2^ and 5 × 10^3^ nm (macropores), OS1 presented the highest pore volumes. On the other hand, OS2 showed the highest pore volume in the range of narrow mesoporosity, which is a consequence of the larger external surface area observed for this ACM (Figure 2).

Table 1 collects the mass surface concentration of phosphorus that was determined by XPS analyses. As can be seen, the phosphorus surface concentration values were found between 2.1 and 4.5%. The presence of phosphorus chemically stable on the surface of the ACMs is a well-known consequence of the activation method with phosphoric acid at certain preparation conditions. This phosphorus remained on the ACM surface, even after the washing step, mainly in form of –C–O–PO_3_ and –C–PO_3_ species [7,9]. In this sense, KL1 showed the highest P surface concentration and also the largest contribution of the more oxidized phosphorus species [22]. The importance of this P content was mainly related to the fact that phosphates and polyphosphate esters that are found on the ACM surface had hydroxyl groups (–OH) that acted as Brönsted acid sites, giving up protons (H+) during the alcohol dehydration reaction [36]. Therefore, there should be a correlation between the amount of phosphorus that remains on the catalyst surface and the acidity of the ACMs.

On the other hand, Bedia et al. analyzed the acidity of different powder activated carbons that were prepared by chemical activation with phosphoric acid by NH_3_-TPD [11]. The total amount of ammonia desorbed showed a clear relationship with the amount of oxygen surface groups that decompose as CO and CO_2_ during the TPD experiments, indicating the relevance of carrying this type of experiments in terms of acidity of the catalysts. Table 1 also summarizes the amount of CO and CO_2_ that evolved from TPD experiments under inert flow. The amount of oxygen surface groups decomposing as CO was significantly higher for KL1, and very similar for AL1 and OS2. The presence of CO-evolving groups was also considerably greater than that of the CO_2_-evolving groups, this latter being related to the small amounts of carboxyl, lactonic, and anhydride surface groups. The higher evolution of CO was mainly observed at temperatures higher than 700 °C and associated with the presence of the C–O–PO_3_ groups that formed during the activation of lignocellulosic materials with phosphoric acid, which decomposed as CO at high temperatures (~860 °C) [7,9,10]. Valero-Romero et al. associated a weak and moderate-strength acidity to these C–O–P type species [12]. Therefore, there seems to be a close relationship between the specific amount of CO-evolving groups and the acidity of these ACMs.

2-propanol decomposition was used to characterize the acidity of the ACMs. Figure 4 shows the steady-state conversions of 2-propanol (IPA) and the selectivity to dehydrogenation and/or dehydration products as a function of the reaction temperature for all of the ACMs (Po_IPA_ = 0.03 atm; W/Fo_IPA_ = 0.1 g·s/μmol). In general, the IPA conversion increased with the reaction temperature and OS2 showed higher catalytic activity than the other ACMs ones at any temperature.

The main reaction product was propylene that derived from the dehydration reaction. The selectivity to propylene was very high at all of the evaluated temperatures. Only small quantities of diisppropyl ether were observed at low temperatures and IPA conversions, showing a selectivity to propylene that was higher than 90% for temperatures higher than 250 °C, for all the ACMs. These results suggested that these ACMs developed a relatively large amount of surface acidity during the preparation procedure and, thus, there would be no need for additional treatments to increase the acidity of these catalysts, such as activated carbon oxidation with nitric acid [37].

Mass transport limitations can be neglected based on the theoretical calculations of Carberry number and the isothermal intraphase internal effectiveness factor, at the operating conditions that were used in this study [12]. The values of apparent rate constant, k, have been obtained using conversion values below 20%, taken into account a global first-order kinetic. From these values and applying the Arrhenius equation, the corresponding values of apparent activation energy (Ea) and the preexponential factor (k_o_) have been calculated for each ACM and they are summarized in Table 2. The values that were obtained for the apparent activation energies were quite similar, ranging from 100 to 130 kJ/mol and of the same order than those that were obtained for different inorganic catalysts (heteropolyacids, zeolites,…) [38,39] and powder activated carbons that were obtained by chemical activation with phosphoric acid from different lignocellulosic waste [11,15].

Preexponential factor values can provide a rough estimation of the amount of active sites per unit of surface area or unit of mass of catalyst. As can be seen, the values that were reported in Table 2 followed the same sequence than the P content that was determined by XPS (see Table 2), KL1 > AL1 > OS2 > OS1, which suggested that the number of active sites are directly related to the presence of phosphates and polyphosphate esters in the different ACMs. A relationship between the amount of surface phosphorus complexes (of C–O–P type) and surface acidity has been also reported [40].

### 3.2. Methanol Decomposition

Once the acidity of the ACMs was evidenced, these catalysts were used for the methanol dehydration reaction. The catalytic activity of the ACMs here reported are denoted in Figure 5, where the methanol conversion, X_MeOH_, as a function of the reaction temperature was represented. The reaction of methanol dehydration was carried out in the presence of air (P_MeOH_ = 0.03 atm, W/F_MeOH_ = 0.01 g·s/µmol). A temperature of 375 °C was selected as the maximum reaction temperature that the carbon monoliths can stand without the significant gasification of the carbonaceous matrix. These carbon-based catalysts could work at such a high temperature without burning out due to the high oxidation resistance that was provided by the presence of these P surface complexes [22]. The highest conversion obtained was approximately 70% at 375 °C, with OS2 being the most active ACM due to its higher value of apparent surface area, which compensates its slightly lower P content. The rest of ACMs follows the sequence OS1>KL1>AL1.

DME was the main reaction product. The selectivity to this product was higher than 90% for all the ACMs up to 350 °C. At 375 °C, the selectivity to DME decreased to around 85% due to the appearance of CO_2_ and, in less extent CO in the outlet stream. The burn-off of the ACMs was considerable at higher temperatures and the methanol dehydration was not evaluated. Other authors have evaluated different activated carbons as catalysts for the methanol dehydration reaction, and they have reported much lower methanol conversions, although under inert atmosphere [5,6]. The P surface complexes C-O-P type play an important role for an efficient and highly selective methanol dehydration to DME, given that they present a weak-to-moderate acid strength [15,16] and provide the carbon surface with high oxidation resistance [10,40]. In a previous work, we had reported that these P surface groups would seem to be the active sites that are responsible for the relatively high conversion of methanol to DME [12]. These active sites had presented a redox function, avoiding (in the presence of air) the catalyst deactivation by coke deposition due to a continuous (re)oxidation of the reduced P surface groups, which are produced by the dehydration reaction, without a significant the burn-off of the carbon matrix of the catalyst.

#### 3.2.1. Stability Study

With the goal of studying the stability of OS2 monolith under reaction, a long-term experiment was carried out at two different temperatures. Figure 6 shows DME yields as a function of the time on stream at 300 and 350 °C, at P_MeOH_ = 0.03 atm and W/F_MeOH_ = 0.1 g·s/µmol. Almost no decrease of the DME yield was observed at 300 °C for 20 h of time on stream, which evidenced the high stability of this catalyst under these operating conditions. At 350 °C, a gradual decrease of the methanol conversion could be observed, which was mainly related to a progressive deactivation of the catalyst by coke deposition. However, it is important to point out that methanol conversions higher than 45% have been obtained for more than 20 h, with selectivity towards DME practically the same, with values higher than 85%.

#### 3.2.2. Influence of Water Vapor in the Reaction Mixture

The methanol dehydration reaction to DME can be significantly influenced by the presence of water vapor, which may shift the reaction equilibrium [41,42]. The influence of the presence of water in the methanol conversion was analyzed for OS2. Figure 7 shows the methanol steady state conversion for different inlet partial pressures of water at 300 and 350 °C in air. For the sake of comparison, methanol equilibrium conversions were also included in the dash lines. Methanol conversions significantly decreased in the presence of water, as compared to the equilibrium conversions, which was due to its competitive adsorption with methanol for the acid sites. Specifically, methanol conversion was reduced by half in the presence of 0.08 atm of water vapor at 350 °C, meanwhile a decrease of almost 70% was observed at 300 °C, which indicated that the influence of water was less pronounced with the increase of reaction temperature, which was probably due to the adsorption enthalpy of water being slightly higher than that of methanol, according to the values that were reported in the literature [43,44]. DME formation also decreased from 99 to 86% in the presence of 0.08 atm of water, at 300 °C.

Valero-Romero et al. reported a methanol conversion decrease from 20 to 11% when the water content in the feed was raised to 0.04 atm at 300 °C, for a powder activated carbon [12]. This decrease was less pronounced than the one that was observed for OS2, in spite of both the powder activated carbon and the monolith (OS2) presenting very similar P contents, derived from XPS analyses. However, the experimental conditions for these two cases were slightly different. The inlet methanol concentration was lower for the case of the powder activated carbon, which would suppose a lower formation of water by the reaction and, therefore, a lower competitive adsorption between methanol and water vapor.

### 3.3. Kinetic Study

The steady-state catalytic methanol conversion data that were obtained at different temperatures were fitted using several kinetic models for an integral reactor. The reactor mass balance equation of the reactor was numerically integrated to calculate the methanol conversion.
(1)$$X=\int_{0}^{\frac{W}{F_{MeOH}{}^{0}}}-r_{MeOH}\cdot d(\frac{W}{F_{MeOH}})$$

In this equation, $-r_{MeOH}$ represented the methanol decomposition rate, while $\frac{W}{F_{MeOH}{}^{0}}$ stood for the methanol space time (g·s/µmol).

The mathematical description of $-r_{MeOH}$ relied on the kinetic mechanism for methanol decomposition over acid catalysts. It has been supposed that methanol dehydration was the prevailing reaction contributing to $-r_{MeOH}$ since selectivity towards dimethyl ether was higher than 90% for most of the experiments. Two different reaction pathways for this reaction were proposed, which involved the associative and dissociative mechanisms [45]. The dissociative mechanism, the most generally accepted, considers that an intermediate methoxyl is formed on the acid site after the adsorption of a methanol molecule [46]. A nucleophilic attack by a second methanol molecule results in the dimethyl ether formation, releasing water as a by-product. In addition, the competitive adsorption of water on the active sites is responsible for a decrease in the catalytic activity. Oxygen is not included in the main reaction pathway, and its role is ascribed to the oxidation of the intermediates that would deliver catalyst deactivation through the formation of light hydrocarbons and coke.

From these considerations, three kinetic models were proposed to describe the methanol decomposition rate: (i) Pseudo-second order (SO) with respect to methanol concentration, (ii) Langmuir-Hinshelwood (LH) mechanism, and (iii) Eley-Rideal mechanism (ER). The resulting kinetic rate expressions are collected in Table 3. The LH and ER mechanisms also include the competitive adsorption of water. It must be noted that the relationship of the kinetic and equilibrium constants, methanol adsorption constant, and water adsorption constant (denoted as *k_sr_*, *K_sr_*, *K_MeOH_,* and *K_H2O_*, respectively) with temperature are described while using Arrhenius and Van’t Hoff laws:(2)$$k=k_{0}\cdot e^{-\frac{Ea}{R\cdot T}}$$
(3)$$K_{i}=K_{0,i}\cdot e^{-\frac{\Delta H_{i}}{R\cdot T}}$$

After selecting the appropriate rate expression, Equation (1) is numerically solved while using an ordinary differential equation solver. The related kinetic and adsorption parameters from Equations (2) and (3) are optimized by minimizing the objective function (OF), which is selected to be the sum of the quadratic difference between the experimental and modelled conversion data divided by the number of experimental data points. The optimization is performed while using a simplex search method. The integration and optimization of the models has been implemented using Matlab^®^.

The optimization results reveal that SO is not able to fairly reproduce the experimental data (OF value three times higher than LH and ER ones, Table 3). Mechanisms considering surface reaction on the active site, such as the ER and LH models, are required in providing an accurate mathematical description of the methanol conversion dependence with temperature on these catalysts. The most probable model seems to be LH, according to the lower value that was obtained for the OF (Table 3), although further experiments at higher methanol pressures would be necessary to confirm this finding.

The modeled data (solid lines) calculated from the optimized LH kinetic and adsorption parameters assuming the integral reactor equation have been also included in Figure 5, showing an excellent agreement. The LH model could describe the impact of the presence of water in the reactor feed on the methanol conversion very well. In this sense, the LH mechanism with competitive water adsorption provides the best model predictions for the methanol conversion in OS2 monolith when different amounts of steam were added to the reactor inlet at 300 and 350 °C (see Figure 7).

Table 4 compiles the values of the LH model parameters. The activation energies that were obtained are in accordance to those previously found for other acid activated carbon catalysts used for methanol dehydration and even, for other inorganic materials [2,6,47]. The most active monolith, OS2, showed the lowest activation energy value. Moreover, the evaluation of the kinetic and adsorption constant at 300 °C (the two last columns on Table 4) revealed that the kinetic constant value followed the order OS1 > AL1 > OS2~KL1, whereas the adsorption constant showed a quite different trend, OS2>KL1~OS1>AL1. The combination of both constant values can explain the higher methanol conversion that was attained at low temperature by the OS1 and OS2 monolith. On the other hand, the lower affinity of OS1 surface for methanol would explain why OS2 outperforms OS1 as the reaction temperature increased. The evaluation of *K_H2O_* value at 300 °C (*K_H2O,300_* = 15.8 atm^−1^) revealed that the affinity of the OS2 surface towards water was higher than that for methanol at those conditions. Therefore, the addition of water to the inlet gas could effectively displace methanol from the adsorption sites when its pressure was high enough, a fact that could be behind the activity drop that was observed for OS2 in the presence of water vapor (Figure 7).

## 4. Conclusions

Activated carbon monoliths (ACMs) from different lignocellulosic biomass waste, such as olive stone (OS), Alcell (AL), and Kraft lignin (KL), were prepared by the direct extrusion of the precursors with phosphoric acid, followed by activation under inert atmosphere, and washing with distilled water. These activated carbon monoliths were used as catalysts for the alcohol dehydration reaction. The highest conversion for 2-propanol decomposition was obtained by the ACMs that were derived from olive stone. Selectivity to propylene was quite high at all of the evaluated temperatures, being higher than 90% from 250 °C, for all the ACMs. The values for the apparent activation energies (supposing a first-order kinetic) are quite similar, ranging from 100 to 130 KJ/mol.

The methanol decomposition reaction was also analyzed under air atmosphere. The highest conversion obtained, without a significant burn-off of the carbonaceous matrix, is approximately 70% at 375 °C, with OS2 being the most active ACM, followed by KL1 and AL1 catalyst. Several kinetic models were evaluated to predict the methanol conversions, while taking also into account the competitive influence of water. The Langmuir-Hinshelwood mechanism, whose rate-limiting step was the surface reaction between two adsorbed methanol molecules, represented the experimental data under the conditions studied very well. An activation energy value of 92 kJ/mol for methanol dehydration reaction and adsorption enthalpies for methanol and water of −12 and −35 kJ/mol, respectively, were obtained.

## Figures and Tables

**Figure 1 materials-12-02394-f001:**
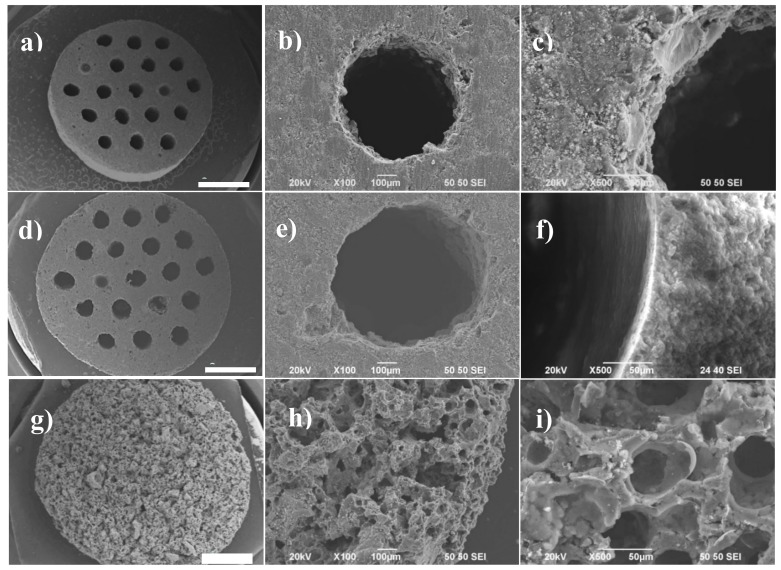
SEM micrographs of the activated carbon monoliths (ACMs) obtained from olive stone (**a**–**c**); Alcell lignin (**d**–**f**); and Kraft lignin (**g**–**i**), at an impregnation ratio of 1. Bar length of (**a**), (**d**), and (**g**) 2 mm.

**Figure 2 materials-12-02394-f002:**
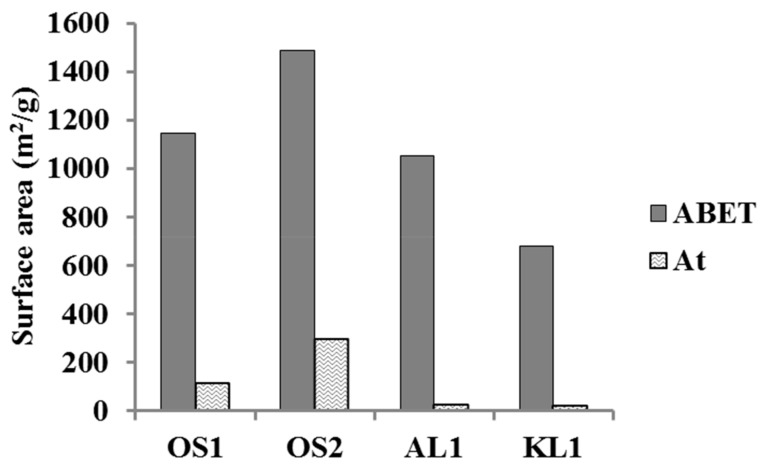
Comparison of the apparent surface area (A_BET_) and external surface area (A_t_) derived from N_2_ adsorption-desorption data of the different ACMs.

**Figure 3 materials-12-02394-f003:**
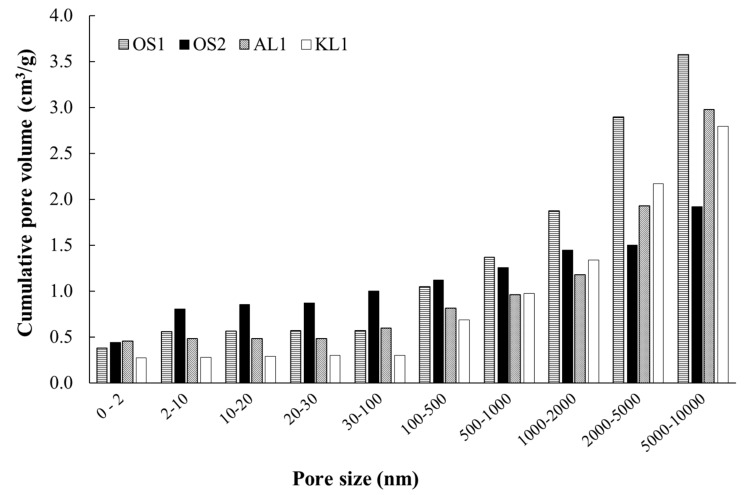
Cumulative pore volume of the ACMs catalysts derived from both the N_2_ adsorption-desorption at −196 °C (micropore and mesopore range) and Hg porosimetry (macropore range).

**Figure 4 materials-12-02394-f004:**
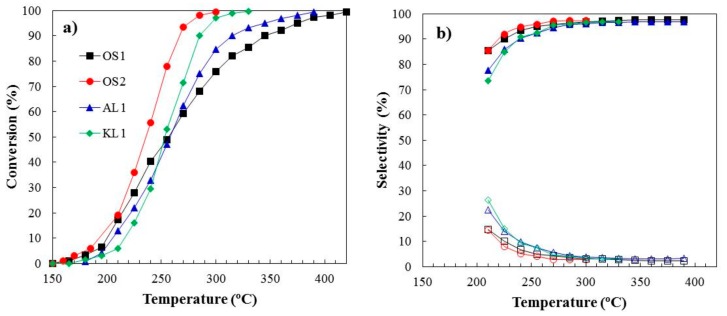
(**a**) 2-Propanol steady state conversion and (**b**) selectivity to propylene (filled marks) and diisoppropyl ether (hollow marks) as a function of the reaction temperature for the different carbon monoliths catalysts under inert atmosphere. P_0IPA_ = 0.03 atm; W/F_0IPA_ = 0.1 g·s/μmol.

**Figure 5 materials-12-02394-f005:**
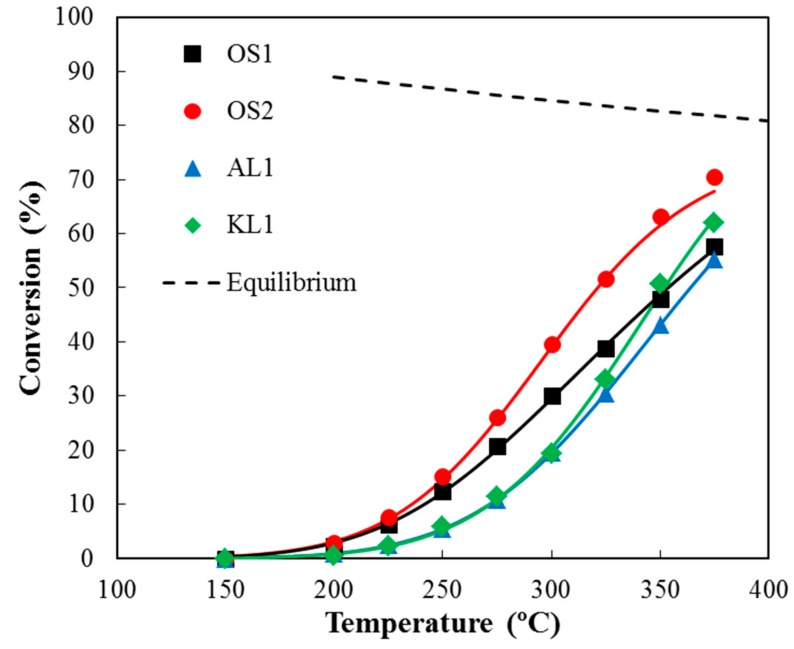
Methanol conversion as a function of the reaction temperature for the different carbon monoliths catalysts under air conditions. P_MeOH_ = 0.03 atm, W/F_MeOH_ = 0.1 g·s/µmol. Dots: experimental data, lines: Langmuir-Hinshelwood model fitting; dash lines: equilibrium conversions.

**Figure 6 materials-12-02394-f006:**
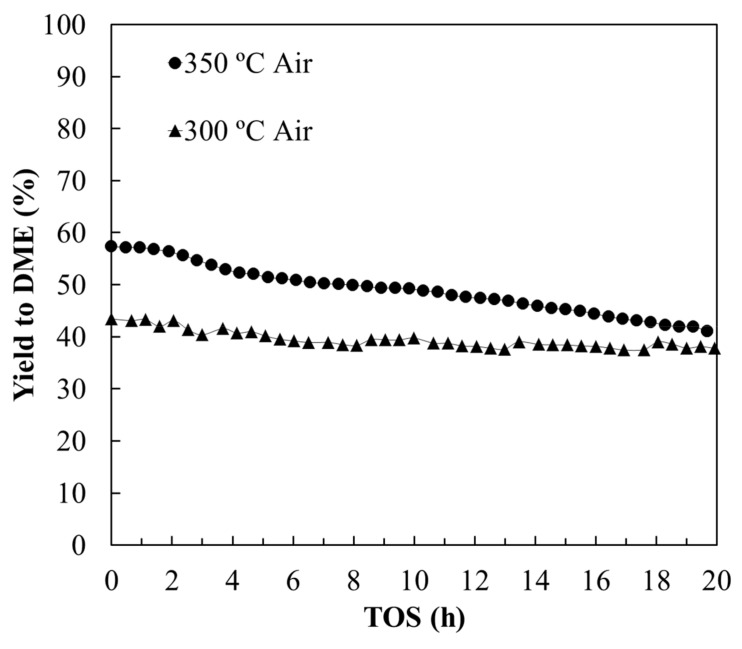
DME yield as a function of the time on stream at different reaction temperatures for OS2 monolith under air atmosphere. P_MeOH_ = 0.03 atm, W/F_MeOH_ = 0.1 g·s/µmol.

**Figure 7 materials-12-02394-f007:**
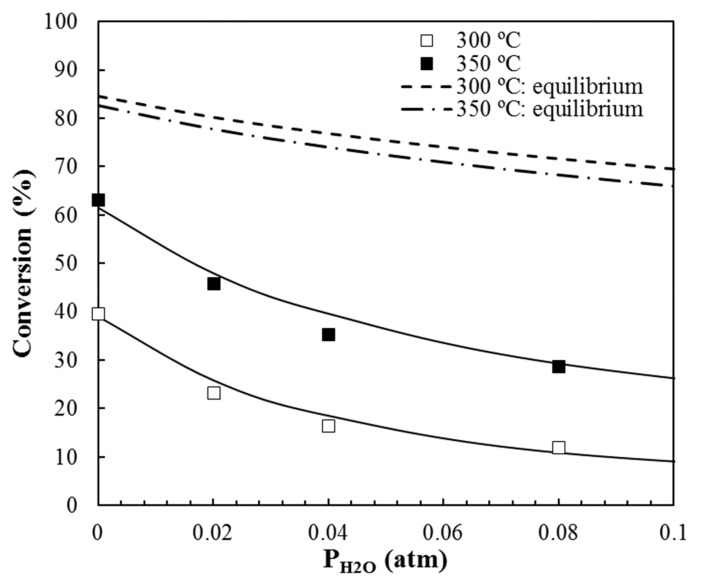
Methanol conversion as a function of different water partial pressures at 300 and 350 °C for OS2 monolith. P_MeOH_ = 0.03 atm, W/F_MeOH_ = 0.1 g·s/µmol. Dots: experimental data, lines: Langmuir-Hinshelwood model fitting; dash lines: methanol equilibrium conversions.

**Table 1 materials-12-02394-t001:** Mass surface concentration of phosphorus determined by X-ray photoelectron spectroscopy (XPS) analyses and CO and CO_2_ evolved quantities from temperature-programmed desorption (TPD) experiments.

ACMs Catalysts	P_XPS_ (wt.%)	CO_TPD_ (mmol/g)	CO_2 TPD_ (mmol/g)
**OS1**	2.8	5.5	0.6
**OS2**	3.2	6.4	0.4
**AL1**	3.7	6.1	0.5
**KL1**	4.5	9.5	0.7

**Table 2 materials-12-02394-t002:** Apparent kinetic parameters for 2-propanol decomposition for all the ACM catalysts.

	Ea (kJ/mol)	ln k_o_
OS1	104	33.14
OS2	116	36.34
AL1	134	39.70
KL1	136	50.99

**Table 3 materials-12-02394-t003:** Kinetic rate expressions for the methanol decomposition.

Model	Rate Expression	OF Values
SO	$-r_{MeOH}=k\cdot P_{MeOH}{}^{2}$	3.387 × 10^−3^
ER	$-r_{MeOH}=\frac{k_{SR}\cdot \left(K_{MeOH}\cdot P_{MeOH}{}^{2}-\frac{P_{DME}\cdot K_{H2O}\cdot P_{H2O}}{K_{SR}}\right)}{1+K_{MeOH}\cdot \;P_{MeOH}+K_{H2O}\cdot P_{H2O}}$	1.025 × 10^−3^
LH	$-r_{MeOH}=\frac{k\cdot \left(K_{MeOH}\cdot P_{MeOH}{}^{2}-\frac{P_{DME}\cdot K_{H2O}\cdot P_{H2O}}{K_{SR}}\right)}{{\left(1+K_{MeOH}\cdot \;P_{MeOH}+K_{H2O}\cdot P_{H2O}\right)}^{2}}$	0.961 × 10^−3^

**Table 4 materials-12-02394-t004:** Langmuir-Hinshelwood (LH) Model parameter values.

Sample	10^−5^·k_0,SR_ mol s^−1^g^−1^	Ea_SR_ kJ mol^−1^	K_0,SR_	ΔH_SR_ kJ mol^−1^	K_0_^MeOH^ atm^−1^	ΔH^MeOH^ kJ mol^−1^	K_0_^H2O^ atm^−1^	ΔH ^H2O^ kJ mol^−1^	10^5^·k_SR,300_ mol s^−1^ g^−1^	K_MeOH,300_ atm^−1^
AL1	49	99	12	21	0.037	−15	0.007	−26	395	0.8
KL1	354	111	43	24	0.025	−18	0.003	−17	269	1.1
OS1	24	96	29	27	0.027	−17	0.011	−33	445	1.1
OS2	6.2	92	19	25	0.155	−12	0.009	−35	276	1.8

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
