# Peer review of "Acid Mesoporous Carbon Monoliths from Lignocellulosic Biomass Waste for Methanol Dehydration"

_materials, 2019, doi:10.3390/ma12152394_

Reviewer 1 Report

The paper Acid mesoporous carbon monoliths from lignocellulosic biomass wastefor methanol dehydration written by Paul O. Ibeh, Francisco J. Garcia-Mateos, Ramiro Ruiz-Rosas, Juana Maria Rosas , Jose Rodriguez-Mirasol and Tomas Cordero may be published in Materials after the minor revision.

This article presents the broad physicochemical characterization of the carbon monoliths materials both Brønsted acid sites. These materials were also described as electrodes by the same authors in their previous paper [Journal of the Taiwan Institute of Chemical Engineers, Volume 97, April 2019, Pages 480-488].

The paper is correct, full description of materials was done and catalytic part together with analysis of kinetics were carried out.

The main remarks concern:

1. In abstract you write: “an activation energy value of 92kJ/mol… was obtained”, activation energy of WHAT?

2. Pore analysis were done by two methods BET and Hg porosimetry. Why two of them? Does Fig.3 present both results or only BET?

3. In 2.1 chapter line 166 – write here a short comment, not only literature [12]

4. I recommend also to calculate the activation energies of methanol conversion together with pre-exponential factors for particular catalysts as the table such as table2 for 2-propanol conversion. Is it the same the order of catalysts acidities obtained by the conversion of two different alcohols?

5. The chapter 3 (Materials and Methods, 3.1, 3.2 and 3.3) should be before Results and Discussion

6. The important remark: LACK OF THE CONCLUSIONS.

Author Response

Reviewer 1

The paper Acid mesoporous carbon monoliths from lignocellulosic biomass wastefor methanol dehydration written by Paul O. Ibeh, Francisco J. Garcia-Mateos, Ramiro Ruiz-Rosas, Juana Maria Rosas, Jose Rodriguez-Mirasol and Tomas Cordero may be published in Materials after the minor revision.

This article presents the broad physicochemical characterization of the carbon monoliths materials both Brønsted acid sites. These materials were also described as electrodes by the same authors in their previous paper [Journal of the Taiwan Institute of Chemical EngineersVolume 97, April 2019, Pages 480-488].

The paper is correct, full description of materials was done and catalytic part together with analysis of kinetics were carried out.

We really appreciate the recommendations and want to thank the Reviewer for the time and effort devoted to improving our manuscript.

We have carefully read all her/his questions. Our response is as follows:

The main remarks concern:

1. In abstract you write: “an activation energy value of 92kJ/mol… was obtained”, activation energy of WHAT?

The value of the activation energy for methanol dehydration reaction has been included in the revised version of the manuscript.

2. Pore analysis were done by two methods BET and Hg porosimetry. Why two of them? Does Fig.3 present both results or only BET?

The porosity of the ACMs was evaluated by N2 adsorption–desorption at −196 °C and by Hg porosimetry. The first technique only evaluates micropores and mesopores, meanwhile macropores are measured by mercury porosimetry. BET equation uses the N2 adsorption isotherm data to calculate an apparent surface area.

In this sense, Figure 3 shows the cumulative pore volume of the ACMs, corresponding to all the micro, meso and macropores range, calculated from both the N2 adsorption-desorption at -196 ºC and Hg porosimetry. The figure caption of Figure 3 has been modified in the revised version of the manuscript in order to clarify this item.

3. In 2.1 chapter line 166 – write here a short comment, not only literature [12]

The following paragraph has been included in the revised version of the manuscript:

“Mass transport limitations can be neglected based on the theoretically calculations of Carberry number and the isothermal intraphase internal effectiveness factor, at the operating conditions used in this study”.

4. I recommend also to calculate the activation energies of methanol conversion together with pre-exponential factors for particular catalysts as the table such as table2 for 2-propanol conversion. Is it the same the order of catalysts acidities obtained by the conversion of two different alcohols?

The data required by the reviewer are reported in the second (pre-exponential factor) and third column (activation energy) of Table 4, and as can be seen, if the logarithm of the pre-exponential factors is carried out, at the same way than the values shown in Table 2, the order of catalysts acidities is quite similar for both alcohols.

5. The chapter 3 (Materials and Methods, 3.1, 3.2 and 3.3) should be before Results and Discussion

We apologize for the misunderstanding, now Materials and Methods section can be found before Results and Discussion section.

6. The important remark: LACK OF THE CONCLUSIONS.

According to the instructions for authors of this journal, Conclusions is an optional section. However, as the reviewer suggests a conclusions section has been included in the revised version of the manuscript.

Reviewer 2 Report

The manuscript reports the use of carbon monoliths to perform the methanol dehydration to DME. The objectives and methods are clear. There are only a few points that would improve the quality of the paper:

- concerning the stability study, the TON (representing the number of mol of DME per mol of active site) would be a better proof of the capability of the catalyst in term of duration (number of cycles it can reach).

- the relationship between the acid sites characterisation and the ranking in methanol conversion is not straightforward and should be better explained. OS2 shows very low values of P (XPS) and CO and CO2 evolutions and it is the most active catalyst... Moreover, it is hard to believe that the kinetic parameters of KL1 for propanol decomposition are the highest: Figure 4 seems to show that OS2 is the most active. Please clarify.

References

Ref 12 has to be completed

Ref 22 has to be completed

Ref 49 has to be completed (Studies in Surface Science and Catalysis Volume 36, 1988, Pages 127-143)

Typos

The authors should check the use of the expression "due to"; I would rather say  "due to the fact that oxygen prevented the acid carbon..." than  "due to oxygen prevented the acid carbon " - See lines 49, 52 and 235.

Line 75:  "being the channels practically indiscernible" should be replaced by "making the channels practically indiscernible" if I understand well.

Figure 1: the scale is not visible for images a, d and g.

Line 111: accumulative should be replaced by cumulative

Line 143: " being this later related" should be replaced by "this latter being related" (if I understand well)

Line 172: the use of ";" is unappropriate

Line 189: "could work a such a" should be replaced by "could work at such a"

Line 196: "Others" should be replaced by "Other"

Author Response

Reviewer 2

The manuscript reports the use of carbon monoliths to perform the methanol dehydration to DME. The objectives and methods are clear. There are only a few points that would improve the quality of the paper:

We really appreciate the recommendations and want to thank the Reviewer for the time and effort devoted to improving our manuscript.

We have carefully read all her/his questions. Our response is as follows:

- concerning the stability study, the TON (representing the number of mol of DME per mol of active site) would be a better proof of the capability of the catalyst in term of duration (number of cycles it can reach). 

We totally agree with the reviewer that TON would be very interesting, but these values were obtained in the reactor operating under continuous conditions and not under different reaction cycles, as the reviewer indicates. For this reason, we have provided conversions data as a function of time on stream. In the new version of the manuscript, we have included the DME yields, which take into account the conversion and the selectivity to DME.

- the relationship between the acid sites characterisation and the ranking in methanol conversion is not straightforward and should be better explained. OS2 shows very low values of P (XPS) and CO and CO2 evolutions and it is the most active catalyst... Moreover, it is hard to believe that the kinetic parameters of KL1 for propanol decomposition are the highest: Figure 4 seems to show that OS2 is the most active. Please clarify.

Although the reviewer points out that OS2 shows very low P contents and CO and CO2 evolution, we would like to highlight that OS2 shows the second largest CO and CO2 evolutions, and a very significant P content of 3.2%, determined by XPS. Moreover, this catalyst shows the highest specific surface area, which plays a very important role in heterogeneous reactions. In this sense, the activity of this catalyst reasonably agrees with the values of these three parameters.

With regard to the second question, the values of apparent rate constant, k, were obtained using conversion values below 20%, in order to avoid any mass and/or heat transport limitations. Effectively, at these lower temperatures, the most active is clearly OS2, as the kinetic parameters values of Table 2 indicate. In this regard, it makes sense that it presents lower activation energy than KL1.

References

Ref 12 has to be completed

Ref 22 has to be completed

Ref 49 has to be completed (Studies in Surface Science and Catalysis Volume 36, 1988, Pages 127-143) 

The following references has been completed in the revised version of the manuscript.

Typos

The authors should check the use of the expression "due to"; I would rather say  "due to the fact that oxygen prevented the acid carbon..." than  "due to oxygen prevented the acid carbon " - See lines 49, 52 and 235.

Line 75:  "being the channels practically indiscernible" should be replaced by "making the channels practically indiscernible" if I understand well.

Line 111: accumulative should be replaced by cumulative

Line 143: " being this later related" should be replaced by "this latter being related" (if I understand well)

Line 172: the use of ";" is unappropriate

Line 189: "could work a such a" should be replaced by "could work asuch a"

Line 196: "Others" should be replaced by "Other"

All the above suggestions of the reviewer have been accomplished in the revised version of the manuscript.

Figure 1: the scale is not visible for images a, d and g.

The length of the bar has been included in the revised version of the manuscript.